# In Vitro Evaluation of Hyperthermia Magnetic Technique Indicating the Best Strategy for Internalization of Magnetic Nanoparticles Applied in Glioblastoma Tumor Cells

**DOI:** 10.3390/pharmaceutics13081219

**Published:** 2021-08-07

**Authors:** Javier B. Mamani, Taylla K. F. Souza, Mariana P. Nucci, Fernando A. Oliveira, Leopoldo P. Nucci, Arielly H. Alves, Gabriel N. A. Rego, Luciana Marti, Lionel F. Gamarra

**Affiliations:** 1Hospital Israelita Albert Einstein, São Paulo 05652-000, SP, Brazil; jbusta25@gmail.com (J.B.M.); taylla.kley@gmail.com (T.K.F.S.); mariana.nucci@hc.fm.usp.br (M.P.N.); fernando.anselmo@einstein.br (F.A.O.); arielly.alves@einstein.br (A.H.A.); gabriel.nery@einstein.br (G.N.A.R.); luciana.marti@einstein.br (L.M.); 2LIM44-Hospital das Clínicas da Faculdade Medicina da Universidade de São Paulo, São Paulo 05403-000, SP, Brazil; 3Centro Universitário do Planalto Central, Brasília 72445-020, DF, Brazil; leopoldo.nucci@gmail.com

**Keywords:** magneto hyperthermia, magnetic nanoparticles, glioblastoma, C6 cells, superparamagnetic iron oxide nanoparticles, AMF, intracellular hyperthermia, extracellular hyperthermia, PLL, static magnetic field, dynamic magnetic field

## Abstract

This in vitro study aims to evaluate the magnetic hyperthermia (MHT) technique and the best strategy for internalization of magnetic nanoparticles coated with aminosilane (SPION_Amine_) in glioblastoma tumor cells. SPION_Amine_ of 50 and 100 nm were used for specific absorption rate (SAR) analysis, performing the MHT with intensities of 50, 150, and 300 Gauss and frequencies varying between 305 and 557 kHz. The internalization strategy was performed using 100, 200, and 300 µgFe/mL of SPION_Amine_, with or without Poly-L-Lysine (PLL) and filter, and with or without static or dynamic magnet field. The cell viability was evaluated after determination of MHT best condition of SPION_Amine_ internalization. The maximum SAR values of SPION_Amine_ (50 nm) and SPION_Amine_ (100 nm) identified were 184.41 W/g and 337.83 W/g, respectively, using a frequency of 557 kHz and intensity of 300 Gauss (≈23.93 kA/m). The best internalization strategy was 100 µgFe/mL of SPION_Amine_ (100 nm) using PLL with filter and dynamic magnet field, submitted to MHT for 40 min at 44 °C. This condition displayed 70.0% decreased in cell viability by flow cytometry and 68.1% by BLI. We can conclude that our study is promising as an antitumor treatment, based on intra- and extracellular MHT effects. The optimization of the nanoparticles internalization process associated with their magnetic characteristics potentiates the extracellular acute and late intracellular effect of MHT achieving greater efficiency in the therapeutic process.

## 1. Introduction

The treatment of cancer remains considered as one of the most challenging health issue. Despite recent, intensive, and rapid advances in new technologies, drugs, and therapies against cancer during the last decades, the glioblastoma multiforme (GBM) is still the most prevalent (14.5% of all tumors and 48.6% of malignant tumors) and the most refractory of the gliomas in adults [1,2]. GMB represents 76% of all gliomas [3], with a high rate of patients deaths (15,000/year in the United States), and less than 10% of patients achieve a 5-year survival rate [4,5]. The main therapeutics approaches for GBM tumors include radiation therapy, chemotherapy, thermal therapy, and surgeries focusing on the best technique of tumor resection; these therapies can be used isolated or combined [6,7]. Regarding the low efficacy of the GBM therapies mentioned above, further studies to improve and/or innovate the therapeutic approach for GBM are required. Nowadays, alternative therapies have been explored, such as cancer immunotherapy, magnetic hyperthermia (MHT), photodynamic therapy, and other treatments that use novel therapeutic agents [8]. The treatments focusing on thermal heating are of great value, as tumor cells have lower tolerance to thermal variance effect compared to healthy cells [6].

In this sense, MHT has been highlighted as a noninvasive approach, which does not require surgery and causes a local effect without damaging healthy cells [2]. This technique has demonstrated good results in different pre-clinical studies with animal models [9,10,11] and also in phase III clinical trials in humans [12,13,14], being promising for treating cancer compared to other heating techniques [15]. This technique involves the administration of MNPs, among them the superparamagnetic iron oxide nanoparticles (SPION), followed by the application of an alternating external magnetic field, which generates heat within the tumor area [16]. The heating produced by MHT remains sustained in temperatures above 43 °C, leading to tumor cell lyses due to low energy dissipation generated by heat associated with the heterogeneity of oxygen supply and nutrient demand caused by excessive sinuous ramifications of blood vessels and absence of lymphatic vessels. This increase in temperature alters the function of many structural and enzymatic proteins in the tumor cells, which, in turn, alters the cellular index of proliferation and induces their death [17,18], as these cells are more sensitive to heat than healthy cells.

The MHT technique is based on two relaxation processes of magnetic nanoparticles—the Néel and the Brownian relaxations, which are associated with intracellular and extracellular MHT processes [19]. For nanoparticles internalized into tumor cells and submitted to an alternating magnetic field (AMF), only the Neel relaxation contributes to thermal energy (intracellular MHT) whereas Brownian relaxation is not relevant, due to the high viscosity of the medium that does not allow the nanoparticles to rotate freely. For nanoparticles non-internalized by cells, the contribution of Néel and Brownian relaxations are relevant for the extracellular MHT process [20].

However, some challenges are involved in the success of this technique such as the use of magnetic nanoparticles (MNPs) that are biocompatible, biodegradable, and with good colloidal stability, as well as the use of MNPs in lower amounts that better generates the local heat capability [8]. The available MHT techniques do not effectively direct the heat to the tumor, therefore currently display low efficiency. Accordingly, this lack of efficiency has led to the development of nanomaterials with properties able to dissipate energy at the tumor site, increasing the efficiency and specificity for deeper tumors [6,21]. In addition, the development of different strategies for the cell internalization process and/or development of nanoparticles with paramagnetic properties [22] as iron oxide nanoparticles also contributes to enhancing this therapy efficiency. 

Nonetheless, the MHT requires an initial efficiency evaluation through in vitro and in vivo studies. To this end, the evaluation of several parameters is required, among them, the type of tumor cell, nanomaterial properties, AMF characteristics, therapeutic temperature, and intra- and extracellular process of MHT [23]. 

The relevance of the in vitro MTH evaluation studies is related to parameters required for envisioning therapeutic strategies for cancer treatment, such as the concentration of magnetic nanoparticles, guidance of the nanomaterial, and time limit application of the therapy, which are all of vital importance to determine the therapy efficiency. In this sense, the in vitro evaluation of parameters involved in the MHT technique can guide the results extrapolation for the development of in vivo studies.

Thus, the aim of this in vitro study was to evaluate the hyperthermia magnetic technique by flow cytometry and bioluminescence (BLI) and the best strategy of internalization of magnetic nanoparticles coated with aminosilane applied by glioblastoma tumor cells, where the internalization strategies were evaluated in the presence of static or dynamic magnetic field, in addition to the use of transfection agent and filtering.

## 2. Materials and Methods

### 2.1. Experimental Design

The experimental design was performed in four phases. The first phase consisted in the specific absorption rate (SAR) process determination of the superparamagnetic iron oxide nanoparticles coated with aminosilane (SPION_Amine_—Figure 1A) using two SPION sizes, two frequencies, and three magnetic field intensities (Figure 1B), and the C6 cells transfection with luciferase (Figure 1C). 

The best SAR value (Figure 1, ¥) determined the SPION used in the second phase, the internalization process of the SPION_Amine_ into the C6_Luc_ cells in three concentrations 100, 200, and 300 µgFe/mL (#) using different strategies (Figure 1D). The strategies of internalization were performed combining the following conditions: cell labeling without MF (Figure 1(D-I)), with static MF (Figure 1(D-II)), with dynamic MF (Figure 1(D-III)), in the absence of Poly-L-Lysine (PLL), and the same conditions in the presence of PLL (IV, V, and VI, respectively), combined with and without filtration (#). The best C6 internalization strategy (Figure 1£), as well as the best frequency and magnetic field intensity (Figure 1B) were performed in a third phase, which consisted of MHT in vitro assays (Figure 1E), following a heating plane (red line) to keep the therapeutic temperature constant. In the last phase, the MHT therapy efficiency evaluation was performed by using the BLI technique and the flow cytometry analysis (Figure 1F). 

### 2.2. Superparamagnetic Iron Oxide Nanoparticles Coated with Aminosilane (SPION_Amine_)

The SPION_Amine_ functionalized with amine group (−NH_2_) is a commercial colloidal solution dispersed in aqueous medium (Chemicell GmbH, Berlin, Germany) at a concentration of 25 mg/mL, density of 1.25 g/cm^3^, with hydrodynamic diameter of 50 nm and 100 nm and number of particles 1.3 × 10^16^/g and 1.8 × 10^15^/g respectively. 

The SPION_Amine_ hydrodynamic diameter value (size of distribution) and zeta potential were measured using the dynamic light scattering (DLS) technique by Zetasizer Nano S system (Malvern, UK). The hydrodynamic size distribution was obtained at 173° manual angle, average of 25 and time of 10 s per medium. Measurements were done on fixed position and at 25 °C with 120 s equilibrium period. 

The mean diameter was obtained by fitting the experimental data to a log-normal distribution function $f(D_{p})=\frac{1}{\sqrt{2\pi }\omega_{P}D_{P}}\exp\left(-\frac{{\left(\ln D_{P}-\ln D_{P}^{0}\right)}^{2}}{2\omega_{P}^{2}}\right)$ with mean diameter $<D_{P}>=D_{P}^{0}\exp(\omega_{P}^{2}/2)$ and standard deviation of mean diameter $\sigma_{P}=D_{P}^{0}{\left[\exp(2\omega_{P}^{2})-\exp(\omega_{P}^{2})\right]}^{1/2}$.

For the zeta potential measures, the samples pH value was determined as 7.4 and kept at 25 °C of temperature.

### 2.3. Magneto Hyperthermia Equipment Configuration

The MHT application used a commercial alternating magnetic field system DM100 (nB nanoScale Biomagnetics), which supports a magnetic field of intensity range from 50 to 300 Gauss and an oscillation frequency range from 305 to 557 kHz. The temperature measures were performed using a fiber optic system (Reflex^TM^, Neoptix) with an accuracy of ±0.3 °C. The MHT system was refrigerated using the minichiller MCA-5-RI (Mecalor, Brazil) system adjusted to a water flow of 2 m³/h and 2.8 bar of pressure, maintaining the water temperature at 12 °C.

### 2.4. SPION_Amine_ Heating Potential

In order to determine the heating potential of nanoparticles, 1 mL of SPION_Amine_ (50 and 100 nm) at a concentration of 10 mgFe/mL dispersed in aqueous medium and thermally insulated in a glass recipient were used, maintaining the room temperature at 20 °C. Heating curves (temperature versus time) of samples were acquired in the following configurations: magnetic field of 50, 150, and 300 Gauss associated with frequencies of 305 and 557 kHz, until achieve the temperature of 55 °C or 200 s of analysis. The heating curves generated were used to calculate SAR values. 

### 2.5. Specific Absorption Rate (SAR) Measurement of SPION_Amine_ in Colloidal Suspensions

SAR reports the heat emitted by SPION_Amine_ per unit of mass exposed to AMF and depend on factors such as magnetic field strength, frequency (f), permeability (μ), and SPION_Amine_ size and shape. The two sizes of the SPION_Amine_ SAR were calculated using the software Zar v1.0 (nanoScale Biomagnetics, Zaragoza, Spain), and the relation $SAR\;\left(W/g\right)=\frac{m_{NP}c_{NP}+m_{1}c_{1}\;}{m_{NP}}\;\left(\frac{dT}{dt}\right)\;max$, where $m_{NP}$ is the nanoparticles mass 25 kg/m³: $m_{1}\;$mass of liquid (1000 kg/m³), $c_{NP}$: specific heat of the nanoparticles (0.16 kCal/kg°C), $c_{1}$: specific heat of the liquid (1.0 kCal/kg°C), and ${\left(dT/dt\right)}_{max}$: maximum gradient of the temperature curve of the colloid submitted to an InH test (K/s).

### 2.6. Cells Culture and Lentiviral Transduction for Luciferase Expression

The cells were cultured in RPMI culture medium (GIBCO^®^ Invitrogen Corporation, CA, USA), supplemented with 10% FBS (GIBCO^®^ Invitrogen Corporation, Carlsbad, CA, USA), and 1% antibiotic–antimycotic solution (GIBCO^®^ Invitrogen Corporation, Carlsbad, CA, USA). The cells were incubated in a humidified atmosphere with 5% CO_2_ at 37 °C (Thermo Fisher Scientific Inc. 3110, Waltham, MA, USA), until achieving the desired density of 75% confluence.

The virion production used for lentiviral transduction was described in a previous study [24]. For lentiviral transduction, 2 × 10^5^ C6 glioma cells per well in a 24-well plate, cultured in 2 mL of RPMI received 10 MOI/cell of the lentiviral vector pMSCV-Luc2-T2A-Pure carrying the pseudotyped viruses of the vesicular stomatitis virus (VSV-G) [25], that codifies the bioluminescent reporter luciferase 2 (viral title: 2.4 × 10^8^ UI/mL), and puromycin resistance gene [26]. To favor the vector entry into C6 cells, we also added polybrene to the culture medium (8 µg polybrene/mL of RPMI). The next day, and after every 2 days we added 1 µg/mL puromycin (Thermo-Fisher Scientific). After two weeks in culture, only the puromycin-resistant cells survived expressing luciferase 2 protein.

### 2.7. Kinetics of Bioluminescent Signal in C6_Luc_ Cells

The kinetic curve of the C6_Luc_ BLI signal was performed to identify the timing of the maximum signal intensity peak and sensitivity detection. For this analysis, we used several cell concentrations/well 10^4^, 5 × 10^4^, 10^5^, 5 × 10^5^, 10^6^, and 3 × 10^6^ in a 48-well plate. The BLI images were acquired by IVIS^®^ Lumina LT Series III equipment (Xenogen Corp., Alameda, CA, USA), after addition of 100 μL of D-luciferin (50 mg/mL) (XenoLight, PerkinElmer). The BLI signal intensity was detected using the automatic exposition time, with f/stop of 4, binning of 8, and the field of view of 12.9 cm with 2 min of interval between each image over 24 h. For BLI image analysis selected regions of interest (ROI) of 2.5 cm^2^ in each well and processed the signal using the Living Image version 4.3.1 software (IVIS Imaging System) in photons/s units.

### 2.8. C6_Luc_ Labeled with SPION_Amine_: Strategies for Internalization

C6_Luc_ labeling with SPION_Amine_ used hydrodynamic diameter with higher SAR value (¥) according to the experiment described in the Section 2.5 as illustrated in Figure 1B.

SPION_Amine_ best internalization strategy was determined using labeling process in different conditions such as in the presence of the static or dynamic magnetic field, using the transfection agent as PLL, and filter (Figure 1D), as also the use of three SPION_Amine_ concentrations. Thus, 10^6^ cells/well plated into a 24-well plate with 2 mL of RPMI medium supplemented with 10% of FBS were cultivate until achieving a confluence of 70%, at 37 °C and 5% CO_2_, in an incubator FormaTM Series II (Thermo Fisher Scientific Inc. 3110, Waltham, MA, USA). Then, three different concentrations of SPION_Amine_—100, 200, and 300 µgFe/mL—were used in the labeling of these cells for 18 h in the following conditions: without magnet, with static magnet positioned below the culture plate a neodymium–iron–boron magnet with 300 mT, or with dynamic magnet which the same magnet positioned below the plate culture, maintained in oscillatory horizontal movement in the frequency of 0.25 Hz, with or without filtration (0.45 μm syringe filter SARSTEDT, Nümbrecht, Germany) (Figure 1(DI–III)), and in the absence or presence of PLL at 0.75 mg/mL, with molecular weight of 300 kDa (Sigma, St. Louis, Mo, USA), due to their electrostatic interaction properties, as shown in Figure 1(DI–VI). The transfection agent complexation (PLL) with SPION for cell labeling is detailed in the Section 2.8.1.

In order to evaluate SPION_Amine_ internalization by C6_Luc_ cells, these cells were washed twice with D-PBS (GIBCO^®^ Invitrogen Corporation, CA, USA), fixed for 2 h in 4% paraformaldehyde, and stained with Prussian blue. The iron position inside the C6_Luc_ cells, were evaluated according to an already established protocol [27,28,29]. Light field images were obtained on the FSX-100 inverted microscope OLYMPUS (Ireland, UK).

#### 2.8.1. Complexation of SPION_Amine_ with Transfection Agent (PLL)

In order to facilitate the SPION_Amine_ internalization into C6, we used a PLL transfection agent. PLL is a positively charged amino acid polymer with a molecular weight of >300 KDa (Sigma, St. Louis, MO, USA), which favors an electrostatic interaction between cells and nanoparticles, mainly in non-phagocytic cells. PLL was complexed to SPION_Amine_ using a concentration of 100 µgFe/mL SPION_Amine_ and 0.75 mg/mL of PLL. The solution was diluted in a final volume 500 μL of RPMI supplemented with 10% FBS. Then, the solution was mixed with a vortex for 5 min and carried to ultrasound (Transonic T460/H—Elma) for 10 min at 37 °C. The zeta potential evaluation of SPION_Amine_ also was performed after the PLL complexation.

### 2.9. Evaluation of the Hydrodynamic Size Distribution and Zeta Potential of Different Colloidal Solutions Using in the Cellular Labeling

The hydrodynamic size distribution and zeta potential were evaluated in the same colloidal solutions used in cellular labeling, such as SPION_Amine_+H_2_O, SPION_Amine_+RPMI, SPION_Amine_+RPMI+Filter, SPION_Amine_+RPMI+PLL, SPION_Amine_+RPMI+PLL+Filter, for which was applied the different strategies of internalization described in Section 2.8.

The samples were prepared with water or RPMI using the concentration of 100 µgFe/mL, with or without PLL transfection agent and filtering or not the solution.

### 2.10. C6_Luc_ Viability Evaluation after Labeling with SPION_Amine_

C6_Luc_ viability was evaluated using the yellow tetrazolium salt (3-(4,5-dimethylthiazol-2-yl)-2,5-diphenyltetrazolium bromide assay (MTT), C6_Luc_ cells were grown in 96-well plates with a fresh culture medium. SPION_Amine_ was added to the cells and maintained in the same condition of labeling described as previously described in the Section 2.8. In addition, 100 μL of MTT reagent was added to the cells (final concentration 1 mg/mL) (Thiazolyl Blue Tetrazolium Bromide; Sigma-Aldrich) and incubated for four hours at 37 °C in 5% CO_2_, followed by the addition of actinomycin D (Sigma-Aldrich) used as a positive control for cell death. The concentration of cell death dose found in this assay was 0.25 μg/mL. After incubation, 100 μL of dimethyl sulfoxide (DMSO Hybri-Max—Sigma-Aldrich) was added to solubilize the formazan crystals formed. Readings were then performed in a DTX 880 Multimode Detector reader (Beckman Coulter) at 490 nm, with subtraction of plate absorbance at 650 nm. Percentage of C6_Luc_ cells viability was calculated using the mean absorbance ratio of the samples triplicates test readings by the absorbance mean of the control wells. Thus, the cell viability is equal to the sample result divided by the control result ×100.

### 2.11. Application of Magneto Hyperthermia in C6_Luc_+SPION_Amine_

After C6*_Luc_* labeling with SPION_Amine_ and the internalization and viability analysis, the best condition for MHT application was selected (Figure 1£). C6_Luc_ cells labeled with SPION_Amine_ (10^6^ cells) resuspended into RPMI (100 μL) were placed in the sample holder and MHT (300 Gauss; 557 kHz) was applied. The MHT application followed the heating planning to maintain a constant temperature of 44 °C for a period of 40 min (Figure 1E).

Sample analyses were performed in triplicate. Samples without MHT exposition were maintained as control samples. The evaluation methods were performed under the following conditions: (I *) C6_Luc_ cells without SPION_Amine_ labeling and without exposure to MHT (C6_Luc_ without MHT), (I) C6_Luc_ cells without SPION and exposed to MHT (C6_Luc_ with MHT); (II *) C6_Luc_ labeled with SPION_Amine_ and not exposed to MHT (C6_Luc_ + SPION_Amine_ without MHT) and (II) C6_Luc_ labeled with SPION_Amine_ and exposed to MHT (C6_Luc_ + SPION_Amine_ with MHT), as shown in Table 1.

### 2.12. MHT Efficiency Evaluation

The evaluation of the MHT efficiency for cell death induction to C6_Luc_ cells was performed by BLI quantification, after a kinetic curve evaluation to find a peak for the maximum intensity signal and by flow cytometry assays using Annexin-V and propidium iodide (PI), both techniques were performed six days after the MHT therapy.

The BLI images of the C6_Luc_ cells labeled with SPION_Amine_ submitted to the MHT process and respective controls (Section 2.11) were performed after addition of D-luciferin (100 μL) per well and acquired in an IVIS Lumina using the same parameters described above (Section 2.7) and respecting the time of the peak of the maximum intensity signal to initiate the images acquisition. The BLI signal was analyzed by the Living Image software in photons/s using the ROI of 2.5 cm^2.^ The cell viability of conditions (I, II, and II *) shown in Table 1 was obtained by a relation between the C6_Luc_ BLI intensity of each condition divided by the C6_Luc_ BLI intensity of condition I * × 100.

The MHT technique efficiency evaluation for C6_Luc_ cells death induction was also performed by flow cytometry. After MHT application, C6_Luc_ cells labeled with SPION_Amine_ and the respective controls (10^6^ cells) were centrifuged, washed in wash buffer, and then cells were labeled with the FITC Annexin-V and PI (Apoptosis Detection Kit I—BD Pharmingen) according to the protocol already established [30]. Briefly, cells were stained with 5 µL of Annexin-V-FITC and 5 µL of PI for 20 min in the dark at room temperature. Next, 400 µL of Annexin-V Binding Buffer was added in each tube and unstained cells and single stains (only Annexin or only PI) were used as FMO control. At least 10,000 events were acquired within 30 min after staining in the LSR-FORTESSA flow cytometer (BD Biosciences, San Jose, CA, USA). The analysis was performed using FlowJo software (BD Biosciences). The cell viability analysis follows the parameters described: viable cells did not stain for Annexin-V or PI (Annexin-V -/ PI -), cells in early stages of apoptosis stained only for Annexin-V (Annexin-V +/ PI -), cells at the latest stages of apoptosis or in necrosis stained for both Annexin-V and PI (Annexin-V +/ PI +). The analysis gates were plotted based on each tube unstained or single stained control.

### 2.13. Statistical Analysis

Data analysis of SAR, cellular viability process by MTT, and MHT technique efficiency by BLI and flow cytometry were presented as the mean and standard deviation. The MHT efficacy evaluation was compared by the ANOVA test, being considered statistical significance for *p* < 0.05. All statistical analysis was performed with the JASP software v.0.14.1 (http://www.jasp-stats.org; Accessed on 10 may 2019).

## 3. Results

### 3.1. Evaluation of the Polydispersion, Heating Curves, and SAR Values of SPION_Amine_

The distribution data of two SPION_Amine_ sizes characterization by DLS were fit to a log-normal distribution function, yielding the mean diameters of 53.8 ± 0.5 nm e 110.1 ± 0.4 nm, displayed in Figure 2A (blue and red curves). These values were compatible with manufacture data for SPION_Amine_ sizes (50 and 100 nm) used in the analysis.

The SPION_Amine_ (50 and 100 nm) displayed heating curves generated by MHT application combining the intensities and frequencies of magnetic fields parameters over time with fast increase of temperature at 150 and 300 Gauss compared to 50 Gauss, independently of the frequencies 305 and 557 kHz (Figure 2B,C). In Figure 2D, the SPION_Amine_ (50 and 100 nm) showed differences in heating curves regarding the best conditions of magnetic field and frequency.

The mean SAR values were higher—184.41 and 337.83 W/g for SPION_Amine_ 50 and 100 nm, respectively—when combining 300 Gauss of intensity with 557 kHz of frequency (Figure 2E,F, red bars). In addition, SAR value of SPION_Amine_ 100 nm was almost double compared to the one of inferior size for these parameters of MHT application, suggesting that the size of SPION_Amine_ has an important influence in heating efficiency. According to these results, 100 nm was defined as an ideal SPION_Amine_ size for the next phase, as represented in Figure 1.

### 3.2. Bioluminescent Kinetics of the C6_Luc_

The BLI signal intensity analysis displayed a maximum peak for BLI cell expression at 20 min, and after achieving the maximum peak of intensity a similar pattern of reduction over time was observed for all cells concentrations tested (Figure 3). The high concentration of cells showed the high amplitude of the signal which was detected only over the concentration of 5 × 10^4^ cells (Figure 3—green triangle curve). The picture inside Figure 3 indicates the cells, which the maximum peak of the BLI signal were inferior of 0.20 × 10^9^ photons/s, embracing the control samples (Figure 3—empty symbols lines) and the concentration of 1 × 10^4^ cells (Figure 3—red solid circle line).

### 3.3. C6_Luc_ Labeled with SPION_Amine_: Strategies of Internalization Analysis

After the selection of 100 nm SPION_Amine_ size according to the highest SAR value, C6_Luc_ cells were labeled with SPION_Amine_, and the prussian blue staining confirmed the *SPION_Amine_* presence in the cell (Figure 4).

The first condition of the labeling process used three concentrations of Fe/mL (Figure 4—#,§,@) and did not use a PLL transfection agent nor filter (Figure 4A,F,K). This condition showed SPION_Amine_ agglomerates in the C6_luc_ cells at 100 μgFe/mL concentration using static magnet field (Figure 4(#F)), as well as in the concentrations of 200 and 300 μgFe/mL using all types of fields, as shown in Figure 4(§A,F,K,@A,F,K), respectively.

The second condition results for all SPION_Amine_ concentrations (Figure 4#,§,@) without PLL and filter (Figure 4B,G,L) showed similar results independent of magnet field used (WM, SM, or DM) and also demonstrated a low loading index of iron in C6_Luc_ labeling comparing to the fourth condition, using PLL during labeling process with filter.

In the third condition, using PLL without filter, the SPION_Amine_ agglomerates result was more evident, as depicted in Figure 4(#C,H,M,§C,H,M,@C,H,M)), and in fourth condition, the PLL with filter decreased the SPION_Amine_ agglomeration in almost all samples, as shown in Figure 4(#D,I,§D,I,N,@D,I,N). In addition, this condition showed that in static magnet field, an increase of SPION_Amine_ agglomeration occurred with its increased concentration, mainly for 200 and 300 μgFe/mL.

The best condition of labeling was observed in 100 μgFe/mL using PLL with filter and dynamic magnet field (Figure 4(#N)), as well as the higher amount of SPION_Amine_ internalized into C6_Luc_ cells.

Therefore, through the optical microscopy image (Prussian blue staining), PLL complexed with SPION_Amine_ allowed better cell labeling compared to its absence, being the internalization increased in the presence of static magnetic field or dynamic magnetic field.

### 3.4. The Hydrodynamic Size Distribution and Zeta Potential of Different Colloidal Solutions Used in the Cellular Labeling

The polydispersion evaluation of SPION_Amine_ hydrodynamic size (nominal value 100 nm) resuspended in different colloidal solutions used in cellular labeling process for internalization strategies (Section 2.8) displayed SPION_Amine_ a hydrodynamic diameter ranging from 107 to 170 nm and a variation in zeta potential from +9.5 to +35 mV (Table 2 and Figure 5).

SPION_Amine_ resuspended in RPMI displayed increased hydrodynamic diameter mean (from 110.2 to 141.1 nm) and a second peak of the mean diameter of 3.60 µm (back curve of Figure 5), which may correspond to a formation of SPION clusters caused by the influence of electrolytes, proteins, and lipids contained in RPMI, as evidenced in the optical microscopy images in the first column of Figure 4.

When this colloidal suspension was filtered, the second peak was eliminated (red curve of Figure 5) and the average diameter (107.2 nm) almost reached the value of SPION_Amine_ resuspended in water (110.2 nm) with a slight change in the zeta potential to +14.2 mV (Table 2), evidenced by optical microscopy images with absence of agglomerations (second column of Figure 4). After the SPION_Amine_ complexation with PLL suspended in RPMI, the average diameter was 169.2 nm with a second peak in 2.5 µm (green curve of Figure 4), evidenced in the optical microscopy images of Figure 4 (third column) that showed higher formation of clusters compared to the labeling in the absence of PLL, and the zeta potential was +20.6 mV. In the last condition, when the filter was applied, the average diameter decreased to 156.6 nm (red curve of Figure 5) and the microscopy images of Figure 4 (fourth column) showed a marked decrease in clusters observed in the unfiltered condition (third column of Figure 4) and the highest zeta potential of +35.0 mV. Therefore, the SPION_Amine_ basal zeta potential measured without PLL was 10.0 ± 1.1 mV and after adding PLL and filter this value achieved 35.0 + 1.9 mV, increasing the efficiency of labeling using PLL during the process (Table 2).

### 3.5. Cellular Viability after Process of C6_Luc_ Labeling with SPION_Amine_

Cellular viability process by MTT (Figure 6) showed more than 94% of the viability of C6_Luc_ labeled with 100 μgFe/mL of SPION_Amine_ in almost all conditions, except for conditions using PLL without filter with static (82%) or dynamic (87%) magnetic field Figure 6A.

C6_Luc_ labeling process with 200 μgFe/mL, the viability was maintained over 98% without PLL (Figure 6B, light gray bars). However, for samples with PLL without filter, the viability was ~80% (Figure 6B, dark gray bars without vertical lines). Similar behavior occurred for 300 μgFe/mL concentration, which showed viability values over 93% for samples without PLL (Figure 6C, light gray bars), and ~75% for samples with PLL and without filter (Figure 6C, dark gray bars without vertical lines).

Accounting the SPION_Amine_ internalization results demonstrated in Figure 4 and cellular viability observed in Figure 6, the best condition of labeling was at 100 μgFe/mL using PLL, filter, and dynamic magnet field with 98% of viability (Figure 6A—dark gray bar with vertical lines of fill) with higher amount of SPION_Amine_ internalized (Figure 4(#N)).

In the three concentrations without PLL (Figure 6A–C, light gray color bars), viability values higher than 94% were observed, but low internalizations were shown by Prussian blue staining (Figure 4(#§@A,F,K) and Figure 4(B,G,L)) compared with images that used PLL (Figure 4(#§@C,H,M) and Figure 4(D,I,N)).

### 3.6. MHT Application and the Technique Efficiency Evaluation

MHT technique application was performed in C6_Luc_ cells with SPION_Amine_ (100 μgFe/mL) in the best condition of labeling. The heating planning measured the temperature values during MHT application (Figure 7A,B), and in order to maintain the heating curves (Figure 7C,D), magnetic field intensities (Figure 7E,F) and frequency of oscillation were controlled (Figure 7G,H).

MHT technique efficiency evaluation was represented by the BLI images (Figure 7I,I*,II,II*) and cellular viability quantifications were demonstrated in Figure 7III. C6_Luc_ submitted to MHT at an intensity of 300 Gauss and frequency of 557 kHz by 40 min (Figure 7I), did not produce an initial increase in temperature, maintaining the environmental temperature (Figure 7C). In addition, this condition did not display differences between BLI images and signals values for C6_Luc_ with or without MHT 4.79 × 10^9^ photons/s (Figure 7I*), resulting in a decrease of only 1% in cellular viability (Figure 7III).

C6_Luc_ labeled with SPION_Amine_ submitted to MHT (Figure 7II) with the same parameters describe above by 40 min (Figure 7B,D,F,H) showed a fast increase in temperature until achieving 44 °C and maintenance of therapeutic temperature, as shown in Figure 7D, resulting in the BLI signal of 1.46 × 10^9^ photons/s and a reduction of 65.6% in cellular viability compared to the control C6_Luc_ labeled with SPION_Amine_ without MHT (Figure 7II*) which the BLI signal was 4.67 × 10^9^ photons/s and the cellular viability was 97.5%.

The cell viability graphic (Figure 7III) showed the viability quantification of C6_Luc_ and C6_Luc_+SPION_Amine_ before and after MHT application using the BLI intensity signal. The groups C6_Luc_ with or without MHT and C6_Luc_ labeled with SPION_Amine_ without MHT showed high viability over time, and the last condition showed low reduction of viability caused by presence of SPION (low toxicity). In contrast, the group labeled with SPION_Amine_ with MHT led to a cellular viability of 31.9% resulting from the tumor cells death (high toxicity). The ANOVA test displayed a significant result for comparison between groups (*p* < 0.001), and this significant result (*p* < 0.001) remains for the comparison between C6_Luc_+SPION_Amine_ with MHT group (Figure 7II) and others (Figure 7I*,I,II*), as demonstrated in the graphic of Figure 7III.

Another cell viability analysis to evaluate the MTH efficacy was the flow cytometry (Figure 8). This figure displays several dot plots, one for each sample. The graph’s quandrants represent the cells conditions, as Q1 (cells stained for Annexin-V- initial apoptosis), Q2 (Cells stained for Annexin-V and PI-late apoptosis or necrosis), Q3 (cells stained only for PI-Necrosis), and Q4 (cells without staining—Viable cells). In accord with this analysis, it was observed that the control sample (C6_Luc_ unlabeled) displayed viability of 99.4% (Figure 8A), MHT application in the cells labeled with SPION_Amine_ and without MHT application (Figure 8B,C) did not interfere in cell viability (99.1% and 97.9%, respectively). However, 69.2% reduction in cell viability was observed when the C6_Luc_ labeled with SPION_Amine_ was submitted to 40 min of MHT at 44 °C, showing cell viability of 30.0% (Figure 8D), also demonstrated in the histogram (Figure 8E). The ANOVA test showed a significant result for group comparison (*p* < 0.001) and this significant result (*p* < 0.001) remains for the comparison between C6_Luc_+SPION_Amine_ with MHT group (Figure 8D) and others (Figure 8A–C), demonstrated in the graphic of Figure 8.

## 4. Discussion

The evaluation of nanomaterial heating efficiency on alternating magnetic field is important for therapeutic application in the tumor treatment by hyperthermia. However, this efficiency depends on the physical-chemical properties of nanoparticles, as size, distribution, magnetic characteristics biocompatibility, and aggregation of nanoparticles are related to the suspension medium and AMF parameters (oscillation frequency and intensity of magnetic field) [31,32]. Thus, the nanomaterial selected for MHT should be analyzed for these properties and parameters aiming application in vitro or in vivo [33,34,35]. The in vitro studies are important to understand the interactions of magnetic nanoparticles with tumor cells, regarding their toxicity, internalization and its influence in the efficacy on intracellular and extracellular magnet hyperthermia.

Regarding the nanomaterial, magnetite-based SPION (Fe_3_O_4_), as selected in the present study, is widely used in several biomedical applications due to its biocompatibility, high magnetic susceptibility, chemical stability, innocuousness, and high saturation magnetization [36]. The functionalization of SPION aims, in particular, to improve cell labeling, which has been demonstrated in studies in vitro with several types of cells, where SPION coated with aminosilane has increasing efficiency of nanoparticles internalization by cells without inducing cytotoxicity [37]. Yet, other strategies are applied to improve the internalization process of nanoparticles in cells.

In this study, we measured the hydrodynamic diameter of SPION_Amine_, and the values found of 53.8 ± 11 nm and 110.1 ± 11 nm were closed to those provided by the manufacturer (50 and 100 nm, respectively) and corroborated other studies that used the same nanoparticles [33,34,35]. On the other hand, the maximum SAR values of SPION_Amine_ (50 nm) and SPION_Amine_ (100 nm) were 184.41 W/g and 337.83 W/g, respectively. Values that are directly proportional to the diameter of the SPION for the range of dimensions involved in this study [38]. Yuan et al., 2011 [35] characterized SPION_Amine_ (100 nm) in AMF with 250 kHz of frequency and 9.12 kA/m of intensity obtaining a SAR of 14,957 W/g. Our calculated SAR value was 337.83 W/g for a frequency of 557 kHz and magnetic field of 300 Gauss (≈23.93 kA/m). A comparison of SAR values is not possible due to the differences in intensity and frequency of magnetic field. However, a comparison can be performed using the intrinsic loss power (ILP) defined by the relation ILP = SAR/(f × H^2^) that takes into account the nanoparticles properties, excluding the influence of the intensity and AMF frequency. Therefore, the ILP of our study was 0.6812 × 10^−8^ Wg^−1^Oe^−2^Hz^−1^ and the value reported of the study by Yuan was 0.4574 × 10^−8^ Wg^−1^Oe^−2^Hz^−1^ [35], representing a small difference in values that can be attributed to the heterogeneity of the coil’s magnetic field. Consequently, our ILP value is according to the values reported in the literature [35]. A SAR value was not found in the literature for 50 nm SPION_Amine._ Thus, based on our own results, and considering that this SAR value should be lower than 100 nm SPION_Amine_, we chose to use the SAR value of 100 nm SPION_Amine_.

As in MHT there are two heat sources depending on nanoparticle location (surrounding tissue or cells and intracellular) [39,40,41], it is important to associate both sources to optimize MHT therapy. In this sense, the present study aims to improve the nanoparticles internalization process in C6 cells using different strategies such as static and dynamic magnetic fields, as well as transfection agent such as PLL, assisted by filtering process of SPION_Amine_, and it was possible to observe that SPION_Amine_ (100 nm) were internalized in C6 cells more efficiently using magnetic field (magnetofection), associated with the transfection agent PLL. Other studies [42,43,44,45,46,47,48] also reported the use of magnetic field aiming at a better internalization of magnetic nanoparticles. Adams et al. [42] studied the internalization of magnetic nanoparticles into cell neurospheres using the static and dynamic magnetic fields and reported increased labeling efficiency over twice when the dynamic magnetic field was used instead static magnetic field, and this enhanced internalization process not influenced cell viability. The process of internalization of SPION_Amine_ (100 nm) by C6 using magnetic fields in our study followed the same principle described by Adams et al. study [42], where the dynamic magnetic field helped the cell labeling process by increasing the nanoparticles contact with cells. The nanomaterial absorption by C6 cells occurs through various forms of cell endocytosis, and in dynamic magnetic field, the vertical magnetic component induces the nanoparticles sedimentation, and the horizontal magnetic component stimulates their endocytosis by cell membrane [42]. Thus, better internalization of SPION_Amine_ (100 nm) by C6 was obtained when the dynamic magnetic field was used, which is based on biological and physical principles described in the literature [49,50].

The transfection agent PLL increased the SPION_Amine_ (100 nm) internalization by cells as shown in Figure 4. This strategy was also evidenced in other studies [51,52,53,54,55,56,57,58,59]. The use of transfection agent as PLL, that is a linear polymer of lysine that bears a positive charge at neutral pH, benefits the electrostatic interaction between nanoparticles and cells [60], as PLL increases the SPION_Amine_ positive charge (increasing the zeta potential) leading to enhanced attraction with C6 cell membrane which is negatively charged. Cells surface negative charge is of great importance in SPION internalization, due to the presence of glycoproteins, glycolipids, and phospholipids in their plasma membrane [61], in particular, the presence of heparan sulfate proteoglycans that are facilitators of endocytosis, providing a higher negative charge in the cell plasma membrane [60]. Thus, zeta potential was measured according to SPION colloidal solution suspension and there was a correlation with important alterations in this potential (from +9.5 to +35.0 mV), and in this way the increasing in positive charge was favorable for internalization process improvement. In this sense, PLL presence enhanced nanoparticles internalization by C6 cells. This type of internalization strategy had already been used in other studies, but in other types of cells, PLL associated with SPION increased its uptake by mesenchymal stem cells [62]; it also increased internalization efficiency in neuronal stem cells [56], umbilical cord mesenchymal stem cells [57], lung cancer cells [58], glioma cancer cells (C6) [54], cancer cells of human prostate (PC3) [53], among others.

In the previously discussed internalization strategies, there is the possibility of the SPIONs clusters formation, due to interactions between nanoparticles covering layer and forces between the nanoparticles such as electrostatic forces, Van der Waals forces, steric forces, and magnetic forces modulated via the Brownian motion of particles [63]. In addition, there are influences of electrolytes, proteins, and lipids contained in biological fluids during the labeling process. This in vitro study demonstrated that these nanoparticle clusters decreased SPION uptake by cells, influencing their cytotoxicity [63,64,65], as evidenced in our results (Figure 6). For in vivo studies, it is necessary to evaluate the SPION stability in several fluids mimicking biological ones and in organ homogenates ensuring the SPION is stable for in vivo best results avoiding unexpected interferences.

After evaluating the best condition of SPION internalization, toxicity, and taking into account the best SAR value, the C6 cells were submitted to MTH process to verify the efficiency of the therapy, influenced by magnetic heating mechanisms (Néel relaxation and Brown relaxation) [66,67,68]. The MHT process can be performed using an alternating magnetic field at high frequency, low frequency, or ultra-low frequency. This study used high frequency, which mainly contributes to intracellular MHT, due to the properties of Neel relaxation, and in the extracellular MHT, both heating mechanisms are involved (Neel and Brown relaxation) [16,41,69,70,71,72]. The results demonstrated decreased cell viability of 70.0% by flow cytometry and 68.1% by BLI which correspond to intracellular and extracellular magneto hyperthermia contribution to cell cytotoxicity. In a previous study performed by this group [73], a reduction of 52% of cell viability was observed in the MHT efficiency evaluation, without the use of any internalization strategy, condition represented in the Figure 4(#A), which can be attributed to MHT extracellular activity due to low efficiency in internalization. Thus, the difference (~16%) in efficacy reduction could be attributed to the contribution of the intracellular MHT process using internalization strategies. This intracellular MHT aspect has been approached in other studies with favorable results. Chen et al. [41] have used the magnetic nanoparticles coated with polystyrene-sulfonic acid internalized into SK-Hep1 hepatocellular carcinoma cells, leading to significant decrease in cell viability. Similarly, it was evaluated in the studies by Lacovite et al. [40] and Shinkai et al. [74] through intracellular MHT in A549 and T-9 cancer cells, respectively, showing the importance of this type of MHT in the therapeutic process. Although these studies have used other types of tumor cells for the application of MHT, we chose to use the C6 cell lineage due to their histopathological and radiobiological similarities with human gliomas, such as nuclear polymorphism, mitosis high rates, tumor necrosis foci, intratumor hemorrhage, invasion of the parenchyma, and neoangiogenesis [75,76], characteristics of aggressive tumors which enable preclinical studies with novel therapeutic approaches as MHT to treat tumors that are considered lethal.

Therefore, the magnetic nanoparticles internalization into tumor cells submitted to MHT generate localized heat, probably influencing these cells’ physiological and biochemical properties, leading to functional and structural alterations driving cell lysis, as the nanoparticles captured by these cells do not undergo Brownian movement which results in decreased heating as reported in previous studies [77,78]. However, in assays with extracellular nanoparticles, the viability is significantly reduced after thermal treatment (MHT), representing the primary acute effect [79].

We can conclude that our study is favorable for antitumor treatment based on intra- and extracellular MHT effects, where the optimization of the internalization process is associated with magnetic nanoparticles characteristics potentialized by the intracellular late effect of MHT added to its extracellular acute effect. Thus, we have achieved high efficiency in the optimization of this therapeutic process.

## Figures and Tables

**Figure 1 pharmaceutics-13-01219-f001:**
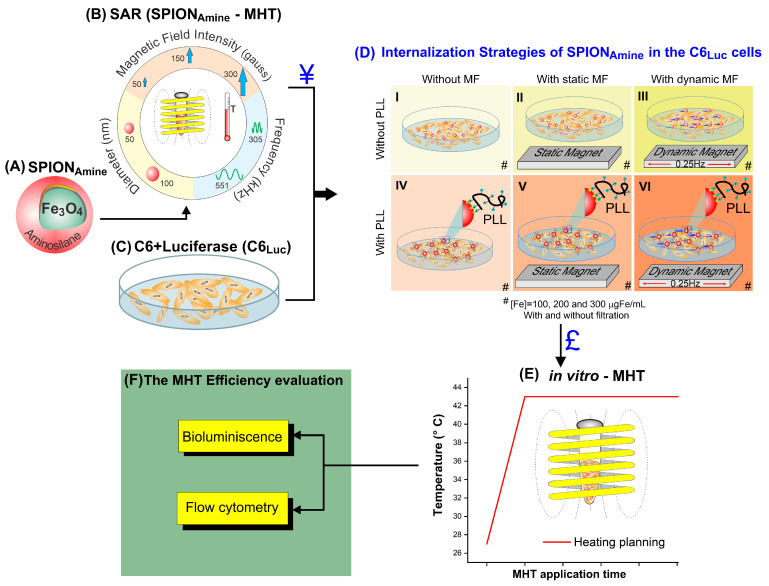
Experimental design. The first stage (**A**–**C**): (**A**) Aminosilane-coated SPION; (**B**) Calculation of the SPION_Amine_ SAR according to magnetic field intensity, frequency, and SPION_Amine_ diameter; (**C**) C6 cells after transfection of luciferase and (¥) Selection of the SPION_Amine_ with the best SAR value. The second stage: (**D**) Internalization strategies (I–VI) of the SPION_Amine_ into the C6 cells in (#) three different concentrations (100, 200, and 300 µgFe/mL), with and without filtration, and also with and without magnetic field (static or dynamic) and combined or not with poly-L-lysine (PLL) transfection agent; (£) Selection the C6 cell labeled with SPION_Amine_ resulting from the best strategy of internalization. The third stage: (**E**) MHT assay in vitro. The fourth stage: *(***F**) evaluation of MHT therapy efficiency.

**Figure 2 pharmaceutics-13-01219-f002:**
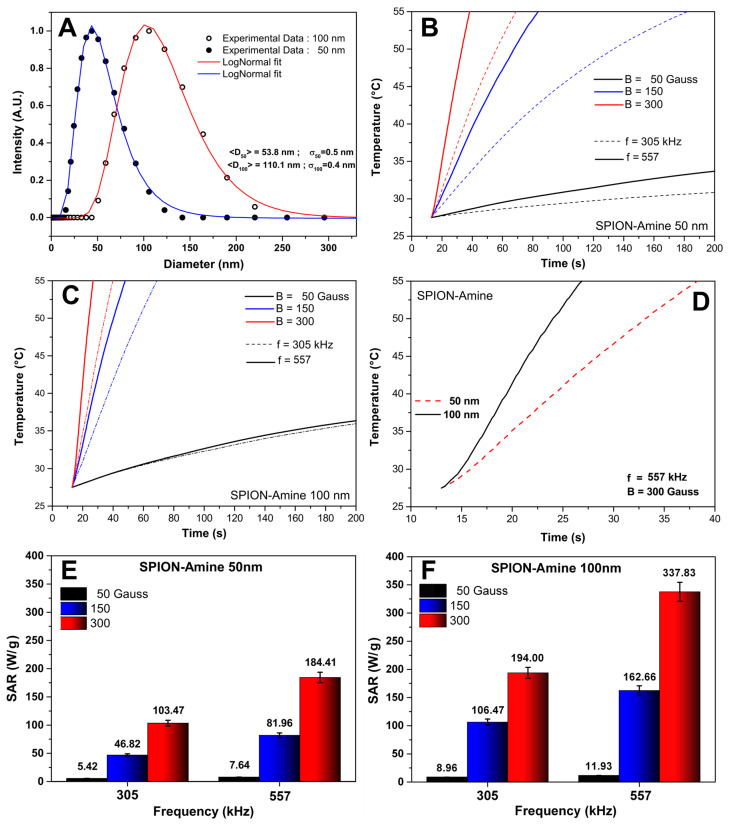
Evaluation of the SPION_Amine_ 50 and 100 nm features for MHT: (**A**) SPION_Amine_ size distribution characterization by DLS, SPION_Amine_ of 50 nm (blue) and 100 nm (red); (**B**,**C**) Heating curves acquired using the following values of magnetic field: 50 Gauss (black), 150 Gauss (blue), and 300 Gauss (red) curves combined with the frequencies 305 kHz (dashed line) and 557 kHz (continuous line); (**D**) Heating curves of SPION_Amine_ of 50 nm (red dashed line) and 100 nm (back continuous line) in the best condition of magnetic field and frequency; (**E**,**F**) Graphic in bars of mean SAR values of the SPION_Amine_ of 50 nm and 100 nm, respectively, analyzing each one of the three parameters of magnetic field with each frequency.

**Figure 3 pharmaceutics-13-01219-f003:**
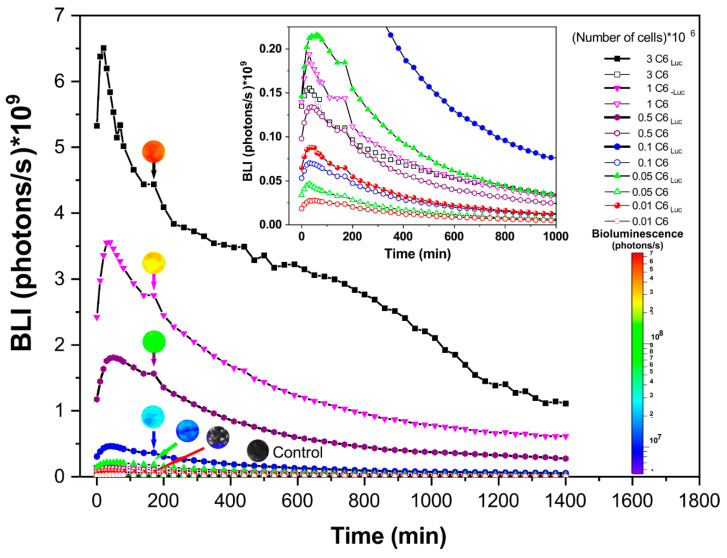
Bioluminescent Kinetics of the C6Luc according to cell concentrations (represented by lines with solid symbols) and the respective controls (represented by lines with empty symbols). The inside figure shows the BLI signal inferior of 0.20 × 109 photons/s. The BLI signal intensity was represented by color score bar (inferior right corner) and the correspondent BLI intensity of the C6Luc cells concentrations and control were represented using a ROI at 180 min.

**Figure 4 pharmaceutics-13-01219-f004:**
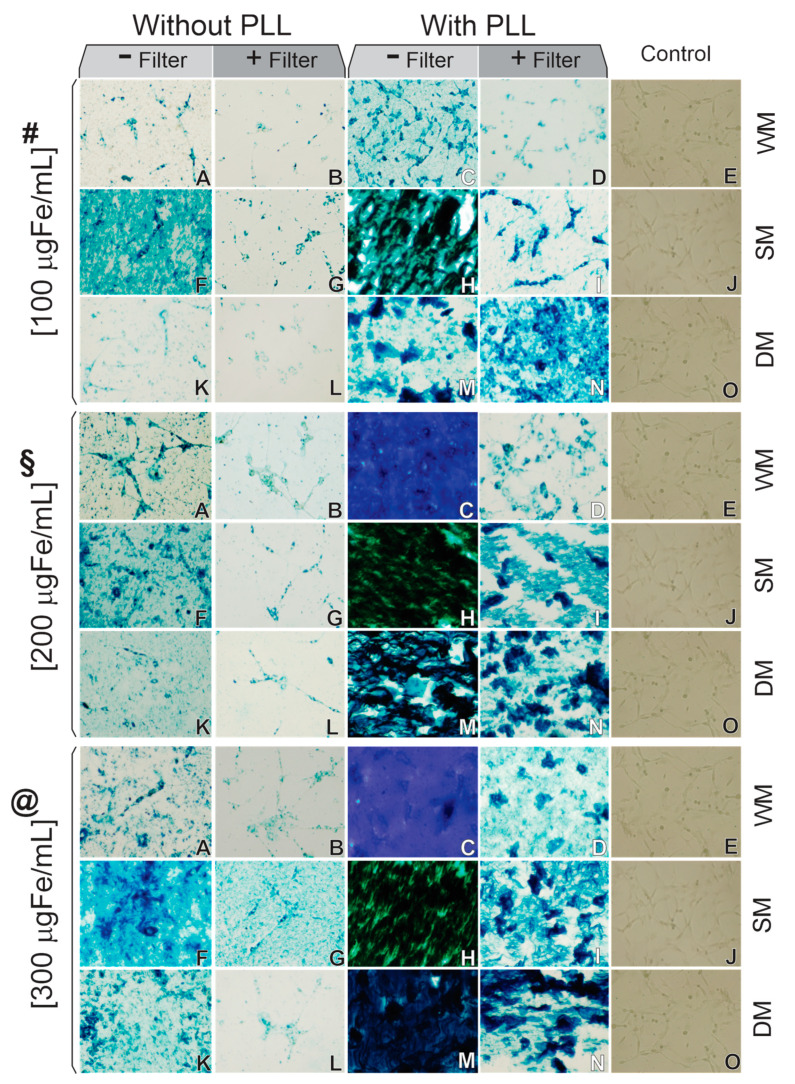
Microscopy optical image of 6_Luc_ labeled with SPION_Amine_ of 100 nm with following strategies of internalization: 100 (#), 200 (§), or 300 (@) μgFe/mL of SPION_Amine_ concentration; without PLL transfection agent and without filter (**A**,**F**,**K**); without PLL and with filter (**B**,**G**,**L**); with PLL and without filter (**C**,**H**,**M**); with PLL and with filter (**D**,**I**,**N**); control conditions (**E**,**J**,**O**); without magnet field (**A**–**E**), with static magnet field (**F**–**J**), and with dynamic magnet field (**K**–**O**). Abbreviations: PLL: Polylysine; WM: without magnet; SM: static magnet; DM: dynamic magnet. The images are shown in 10× magnification.

**Figure 5 pharmaceutics-13-01219-f005:**
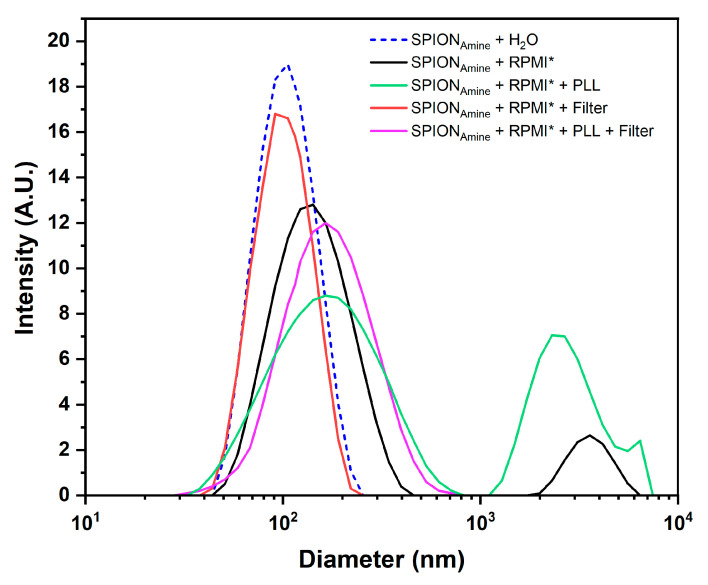
Evaluation of the SPION_Amine_ polydispersity resuspended in different colloidal solutions used in cellular labeling, in the presence of the transecting agent and the filtering process by the technique of dynamic light scattering. * RPMI culture medium supplemented with 10% fetal bovine serum.

**Figure 6 pharmaceutics-13-01219-f006:**
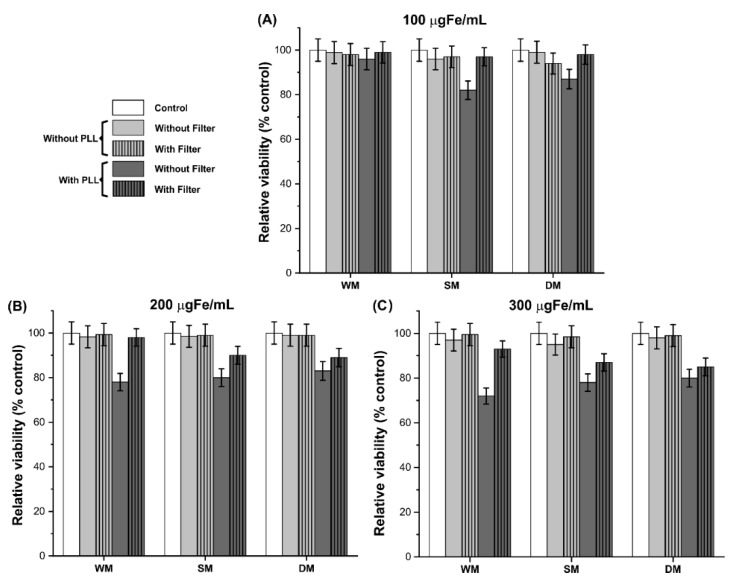
MTT assay for cellular viability evaluation after process of C6_Luc_ labeling with SPION_Amine_ for concentrations: of 100 (**A**), 200 (**B**) and 300 μgFe/mL (**C**). Control conditions are demonstrated by the white bars, without PLL condition in light grey bars and with PLL condition in dark grey bars. The conditions that combined the use of filter were represented by bars with fill of vertical lines. Abbreviations: WM: without magnet; SM: static magnet; DM: dynamic magnet.

**Figure 7 pharmaceutics-13-01219-f007:**
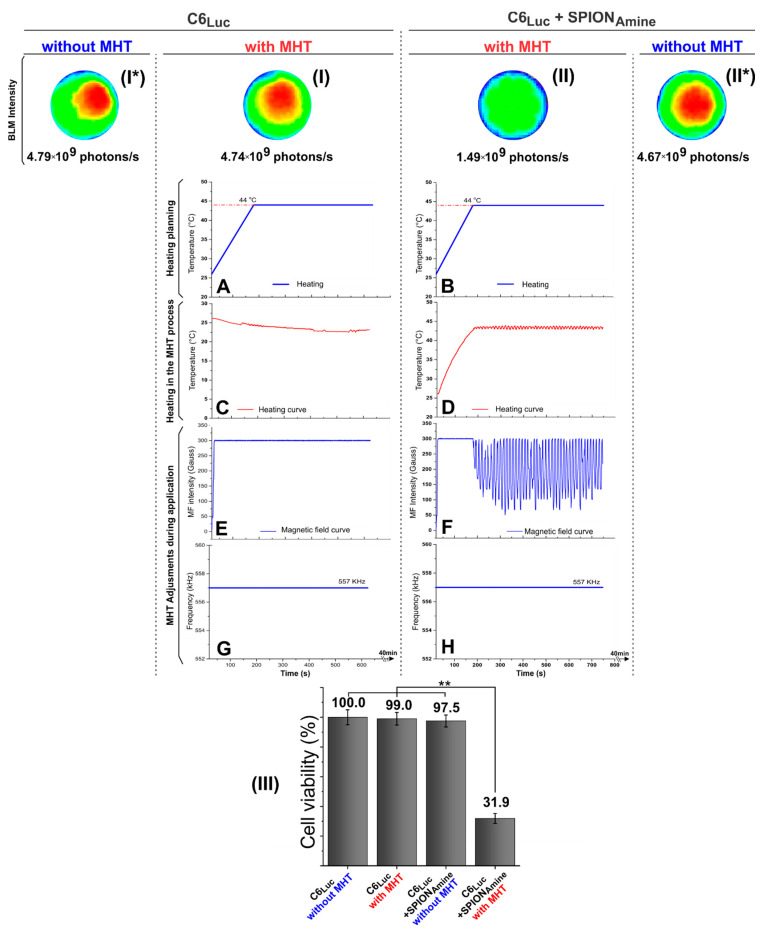
MHT technique efficiency evaluation. (**I**) C6_Luc_ submitted to MHT; (I*) C6_Luc_ did not submit to MHT; (**II**) C6_Luc_ labeled with 100 μgFe/mL SPION_Amine_ submitted to MHT; (**II***) C6_Luc_ labeled with 100 μgFe/mL SPION_Amine_ did not submit to MHT; (**III**) the cellular viability quantification using the BLI intensity signal for each condition. MHT applications in C6_Luc_ and in C6_Luc_ labeled with SPION_Amine_, where (**A**,**B**) heating planning curve, (**C**,**D**) C6_Luc_ heating curve, (**E**,**F**) Magnetic field curve, (**G**,**H**) frequency of field during MHT. ** *p* < 0.001 compared with (**II**) group. Abbreviations: MHT: Magnetic hyperthermia therapy, Luc: Luciferase, min: minutes, SPION: superparamagnetic iron oxide nanoparticles, MF: magnetic field.

**Figure 8 pharmaceutics-13-01219-f008:**
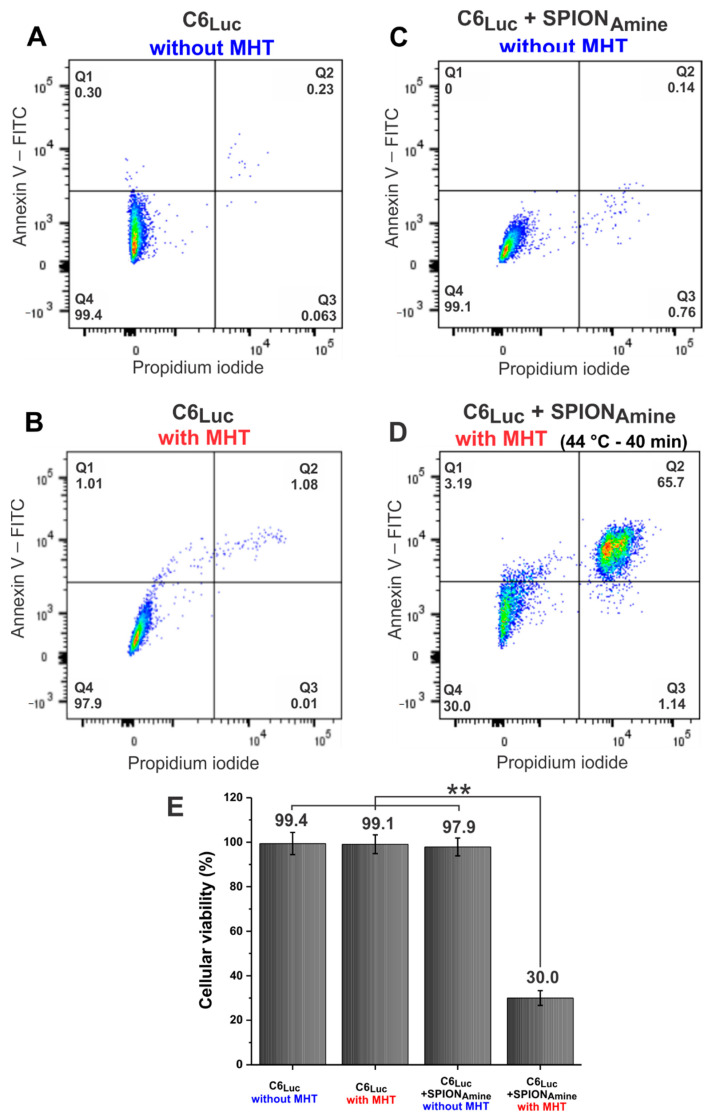
C6_Luc_ cells viability analysis by flow cytometry. (**A**) C6_Luc_ without MHT application; (**B**) C6_Luc_ submitted to MHT; (**C**) C6_Luc_ labeled with 100 μgFe/mL SPION_Amine_ without MHT application; (**D**) C6_Luc_ labeled with 100 μgFe/mL SPION_Amine_ submitted to MHT at 44 °C; and (**E**) cell viability in each experimental group. ** *p* < 0.001 compared with (**D**) group. Abbreviations: MHT: Magnetic hyperthermia therapy, Luc: Luciferase, min: minutes, SPION: superparamagnetic iron oxide nanoparticles, FITC: Fluorescein isothiocyanate. Q1: early necrotic cells (AnxV+/PI-); Q2: necrotic or late apoptotic cells (AnxV+/PI+); Q3: necrotic cells (AnxV-/PI+); Q4: viable cells (AnxV-/PI-).

**Table 1 pharmaceutics-13-01219-t001:** Conditions for MHT in vitro application.

Condition	C6_Luc_	SPION_Amine_	MHT
**I ***	+	-	-
**I**	+	-	+
**II ***	+	+	-
**II**	+	+	+

I *, II * controls samples without MHT exposition.

**Table 2 pharmaceutics-13-01219-t002:** Hydrodynamic diameter values around 100 nm obtained from the polydispersion graphs in each condition of SPION_Amine_ colloidal suspension used in the cell labeling process.

Conditions	Hydrodynamic Diameter (nm)	Zeta Potential (mV)
SPION_Amine_+H_2_O	110.2 ± 0.4	+9.5 ± 1.0
SPION_Amine_+RPMI	141.1 ± 0.5	+10.0 ± 1.1
SPION_Amine_+RPMI+Filter	107.2 ± 0.3	+14.2 ± 1.6
SPION_Amine_+RPMI+PLL	169.2 ± 0.6	+20.6 ± 1.2
SPION_Amine_+RPMI+PLL+Filter	156.6 ± 0.6	+35.0 ± 1.9

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
