# Peer review of "In Vitro Evaluation of Hyperthermia Magnetic Technique Indicating the Best Strategy for Internalization of Magnetic Nanoparticles Applied in Glioblastoma Tumor Cells"

_pharmaceutics, 2021, doi:10.3390/pharmaceutics13081219_

Round 1

Reviewer 1 Report

The article entitled “In vitro evaluation of the hyperthermia magnetic technique based on the search for the best strategy of internalization of magnetic nanoparticles coated with aminosilane applied in glioblastoma tumor cells” presents the evaluation of the magnetic hyperthermia (MHT) technique and the best strategy for internalization of magnetic nanoparticles coated with aminosilane (SPIONAmine) in glioblastoma tumor cells. In my opinion, the article sounds repeated with many previous articles using a similar design by the authors, in the MDPI as well. However, some suggestions are given below to improve the quality.

  1. The title is too long; I suggest modifying it.
  2. The introduction sounds a bit confusing, as the authors discussed all therapies and the importance of materials in the first paragraph, later in the next paragraph started again discussing MHT and then particles. I would suggest making it logically interesting.
  3. Add more details of the study performed at the end of the introduction.
  4. Discuss the materials used in the experimental section. Moreover, the methodology of functionalization should be explained clearly so as to reproduce the same by others.
  5. The authors did not apply any statistical analysis even in bioefficacy studies, suggest using and provide the details in the experimental section.
  6. I suggest authors discuss Intra and extracellular MHT effects frequency range and the role of superparamagnetic iron oxide nanoparticles acute effect on tumors?
  7. Check symbols and remove unwanted symbols? Check subscripts and superscripts? Also, check abbreviations some abbreviations are written wrong? Check grammar and spelling mistakes?

Author Response

Reviewer #1

The article entitled “In vitro evaluation of the hyperthermia magnetic technique based on the search for the best strategy of internalization of magnetic nanoparticles coated with aminosilane applied in glioblastoma tumor cells” presents the evaluation of the magnetic hyperthermia (MHT) technique and the best strategy for internalization of magnetic nanoparticles coated with aminosilane (SPIONAmine) in glioblastoma tumor cells. In my opinion, the article sounds repeated with many previous articles using a similar design by the authors, in the MDPI as well. However, some suggestions are given below to improve the quality.

Answer: This article does not have the same experimental design as previously published articles. The article published in MDPI mentioned by the reviewer evaluated an in vivo study the efficiency of the MHT process in multiple applications using PET/CT, bioluminescence techniques, and behavioral assessment, highlighting the importance of several applications of MHT in the functional, behavioral, and reduction tumor volumetric aspects.

On the other hand, the objective of this in vitro study is different, an aim to evaluate the hyperthermia magnetic technique best strategy of internalization of magnetic nanoparticles coated with aminosilane applied to glioblastoma tumor cells, where the internalization strategies were evaluated in presence of the static or dynamic magnetic field, using transfection agent and filtering. As also, highlighting aspects of intra and extracellular MHT what was not approached in the  previous manuscript.

Therefore, we can state that the present study has different objectives from previous studies. The similarities are only in the assessment of SAR in both studies, as well as the labeling process, but without the use of any internalization strategy (no magnetic field, absence of transfection agent, and no filter). However, these assessments are common in most MHT studies in the literature.

  1. The title is too long; I suggest modifying it.

Answer:  Thank you for your observation. The title was modified as requested.

  1. The introduction sounds a bit confusing, as the authors discussed all therapies and the importance of materials in the first paragraph, later in the next paragraph started again discussing MHT and then particles. I would suggest making it logically interesting. Answer: Thank you for your observation. The introduction has been reorganized as requested, improving  the information comprehension.

  1. Add more details of the study performed at the end of the introduction.

Answer:  Thank you for your observation. As requested, a detailed description of the study was placed at the end of the introduction.

  1. Discuss the materials used in the experimental section. Moreover, the methodology of functionalization should be explained clearly so as to reproduce the same by others.

Answer:  Thank you for your observation. We added in the discussion section of the manuscript more information about the materials used in the present study. In the method section of the manuscript, we have detailed the material description. Regarding the SPION functionalization with aminosilane, as is a commercial colloidal solution, this information was added in the manuscript, but the transfection agent (PLL) complexed with SPION for cell labeling was described in more detail in item 2.8.1.  (New item)

  1. The authors did not apply any statistical analysis even in bioefficacy studies, suggest using and provide the details in the experimental section.

Answer:  Thank you for your observation. The statistical analysis item was added in the method section of the manuscript and the respective analyses of MHT efficiency were added in the result sections of the manuscript.

  1. I suggest authors discuss Intra and extracellular MHT effects frequency range and the role of superparamagnetic iron oxide nanoparticles acute effect on tumors?

Answer:  We have added more information in the discussion section of the manuscript.

  1. Check symbols and remove unwanted symbols? Check subscripts and superscripts? Also, check abbreviations some abbreviations are written wrong? Check grammar and spelling mistakes?

Answer:  Thank you for your observation. We have  checked in the manuscript all the points suggested by the reviewer .

Reviewer 2 Report

The submitted manuscript concerns itself with studying the magnetic hypothermia as a technique in the case of certain magnetic particles (coated superparamagnetic iron oxide nanoparticles).

The study is realized under varying conditions (2 particle sizes x 2 frequencies x 3 field intensities), which allows for a fair comparison of the effects of these parameters. 
The design of the experiment is pretty complex, and the preparation of the samples is rather intricate, but the experimenters took their time to verify the particle diameter distributions and perform flow citometry. 
The number of sample combinations and conditions is impressive, but while the optical microscopy figure is interesting, the interpretation is a bit lacking here. For example, which pictures (samples) have interesting features, from the point of view of the experimenters? A few comparisons are indeed drawn, but there doesn't seem to be someting striking. Also, the image letters seem to be wrong here (probably a formatting issue), so it's even harder to figure out what's going on.

The third and fourth phase are even more interesting, since the experiments were performed in vitro, and a therapeutic assessment, using a variety of techniques. Still, the presentation is confusing at some points, since the authors  amalgamate different types of results in the same figure, resulting in blocks of figures, which taking into account the accompanying text, it is hard to follow.

There's no doubt that the results are there - and in fact the underlying experimental work was probably massive - I think there are probably several papers' worth of good results, but the way it is presented might not make readers see the point the authors are trying to make throughout the paper.

In addition, what could be an improvement is if the authors would specify why the C6 cells were used (and not another type of cell) - or viceversa, why choose the SPION particles to use in these types of cells.

Overall, the paper is good - experimentally – and the results are explained in a decent manner. A bit of effort is needed on the presentation part, and maybe even in justification and interpretation of some of the results.

I consider that the work is decent and adequate for the journal "Pharmaceutics". Considering the results are original, and the work is very extensive, I believe that the paper could be published in the aforementioned journal, provided the authors can address the above observations and suggestions.

Author Response

Reviewer #2

The submitted manuscript concerns itself with studying the magnetic hypothermia as a technique in the case of certain magnetic particles (coated superparamagnetic iron oxide nanoparticles). The study is realized under varying conditions (2 particle sizes x 2 frequencies x 3 field intensities), which allows for a fair comparison of the effects of these parameters. The design of the experiment is pretty complex, and the preparation of the samples is rather intricate, but the experimenters took their time to verify the particle diameter distributions and perform flow citometry.

The number of sample combinations and conditions is impressive, but while the optical microscopy figure is interesting, the interpretation is a bit lacking here. For example, which pictures (samples) have interesting features, from the point of view of the experimenters? A few comparisons are indeed drawn, but there doesn't seem to be someting striking. Also, the image letters seem to be wrong here (probably a formatting issue), so it's even harder to figure out what's going on.

Answer: The following changes were made for better understanding of this study:

  • Figure 1 was redrawn for a better understanding of the strategies used for internalization of SPION in the tumor cells;
  • New items were added in the methods and results section, improving the organization of the study stages and providing more clarity of the results found;
  • A complete new experiment was performed and added for SPION polydispersion in different colloidal suspension media, as well as their measurement for zeta potential. These new results, assisted the interpretation of internalization strategies findings;
  • The process of complexing SPION with the transfection agent was described in detail in the method section;
  • The statistical analysis item was added to the method session and the respective analysis in the results
  • Some figures have been modified to improve the manuscript comprehention;
  • The description of results and discussion has been improved, by adding more details.

The third and fourth phase are even more interesting, since the experiments were performed in vitro, and a therapeutic assessment, using a variety of techniques. Still, the presentation is confusing at some points, since the authors  amalgamate different types of results in the same figure, resulting in blocks of figures, which taking into account the accompanying text, it is hard to follow.

Answer: Thanks for your suggestion. We have modified the text and Figures to improve the comprehension of the manuscript.

There's no doubt that the results are there - and in fact the underlying experimental work was probably massive - I think there are probably several papers' worth of good results, but the way it is presented might not make readers see the point the authors are trying to make throughout the paper.

Answer: Thank you for your suggestion. We have modified the text and added  more details as mentioned in the first question.

In addition, what could be an improvement is if the authors would specify why the C6 cells were used (and not another type of cell) - or viceversa, why choose the SPION particles to use in these types of cells.

Answer: Thank you for your observation. We have added a explanation about the use of C6 cells and about the choice for SPION in the discussion section of the manuscript.

Overall, the paper is good - experimentally – and the results are explained in a decent manner. A bit of effort is needed on the presentation part, and maybe even in justification and interpretation of some of the results.

Answer: Thanks for your suggestion. We have improved the the results explanations and added more details to the text, and also linked the results topics in order to  enrich the interpretation of the manuscript findings.

I consider that the work is decent and adequate for the journal "Pharmaceutics". Considering the results are original, and the work is very extensive, I believe that the paper could be published in the aforementioned journal, provided the authors can address the above observations and suggestions.

Reviewer 3 Report

In the present manuscript, the authors report the development of aminosilane-coated SPIONs for the application in tumor hyperthermia. Nevertheless, several points need to be addressed before this manuscript can be considered for publication.

The manuscript text must be carefully revised.

The materials and methods section must be clarified.

The concentration used in the heating potential studies does not correlate with that explored in the cell studies (10mg/mL vs 300 µg/mL). In that way, it is difficult to conclude about the real therapeutic potential of the SPIONS.

The authors must evaluate the nanoparticles’ cytocompatibility at different time points. Please have in mind that reductions in cell viability superior to 30% indicate that the materials are cytotoxic.

The nanoparticles’ colloidal stability must be characterized in different media. Aggregation phenomena significantly impact cellular uptake.

The utilization of poly-L-lysine is not clear. The PLL was used as coating of aminosilane-coated SPIONS or as a pre-treatment for the cell cultures before the uptake studies.

If the PLL was used as a coating the authors must study the stability of this “functionalization”, determine the amount of PLL adsorbed in the SPIONS, and the possible changes in the size and charge of the SPIONS.

A similar concern arises regarding the filter conditions in cell studies. The SPIONS were submitted to filtration before the studies? If so, how this affects the administered dose? It would be important to present DLS size distribution graphics before and after filtration.

Figure 4, scale bars are missing.

Figure 6, the graphics A-H are difficult to read. Which concentration of SPIONS were used in this study?

Figure 7, cytometry data shows a bad compensation/calibration of PI and Annexin V channels.

The authors state that the PLL increase the SPIONS uptake by increasing its positive charge, but how the poly-L-lysine interacts with the SPIONS to mediate its uptake on the cells?

Author Response

Reviewer #3

In the present manuscript, the authors report the development of aminosilane-coated SPIONs for the application in tumor hyperthermia. Nevertheless, several points need to be addressed before this manuscript can be considered for publication.

  1. The manuscript text must be carefully revised.

Answer:  Thank you for your suggestion. We carefully reviewed the manuscript, modifying some figures and text to improve the comprehension

Answer: The following changes were made for better understanding of this study:

  • Figure 1 was redrawn for a better understanding of the strategies used for internalization of SPION in the tumor cells;
  • New items were added in the methods and results section, improving the organization of the study stages and providing more clarity of the results found;
  • A complete new experiment was performed and added for SPION polydispersion in different colloidal suspension media, as well as their measurement for zeta potential. These new results, assisted the interpretation of internalization strategies findings;
  • The process of complexing SPION with the transfection agent was described in detail in the method section;
  • The statistical analysis item was added to the method session and the respective analysis in the results
  • Some figures have been modified to improve the manuscript comprehention;
  • The description of results and discussion has been improved, by adding more details.

  1. The concentration used in the heating potential studies does not correlate with that explored in the cell studies (10mg/mL vs 300 µg/mL). In that way, it is difficult to conclude about the real therapeutic potential of the SPIONS.

Answer:  Thank you for your observation. The therapeutic SPION potential is unaffected by the different concentrations used (the SAR remains constant), which just changes is the application time of the therapeutic process (Figure 2C and 7D). The study on heating potential (10mg/mL of SPIONAmine with 100nm) represented by Figure 2C shows a heating curve of AMF using 300 gauss and 557 kHz of AMF, in which was observed that at ~20 seconds the temperature achieved 44oC (Therapeutic temperature planned). Regarding to MHT efficiency evaluation of the C6Luc labeled with SPIONAmine (100µg/mL and 100nm) with the same parameters of AMF (frequency and magnetic field), the curve of the therapeutic process achieved to temperature of 44oC in ~200 seconds and this temperature of therapeutic process remain during 40min.

  1. The authors must evaluate the nanoparticles’ cytocompatibility at different time points. Please have in mind that reductions in cell viability superior to 30% indicate that the materials are cytotoxic.

Answer:  Thank you for your observation. We consider it important to carry out the evaluation at different times and thus be sure of the biocompatibility of the nanomaterial. In this sense, the toxicity of the different conditions was evaluated by MTT right after the labeling process, whereas by BLI and cytometry this evaluation occurred 6 days after the application of hyperthermia magnet therapy. During this period, both the first measurement by MTT and the later by BLI and cytometry did not show toxicity greater than 6%. Therefore, we can say that the SPIONAmine has a low toxicity.

  1. The nanoparticles’ colloidal stability must be characterized in different media. Aggregation phenomena significantly impact cellular uptake.

Answer:  Thank you for your suggestion. We performed a new experiment evaluating the SPION colloidal stability added in the method section, the item 2.9 – “Evaluation of the hydrodynamic size distribution and zeta potential of different colloidal solutions used in the cellular labeling”, being additionally measure the zeta potential in the same conditions. We also added one new figure and table with the respective results in the item 3.4 of the manuscript.

  1. The utilization of poly-L-lysine is not clear. The PLL was used as coating of aminosilane-coated SPIONS or as a pre-treatment for the cell cultures before the uptake studies.

Answer:  Thank you for your observation. We added an item in the methods section explaining the purpose of using the PLL, as well as the procedure for complexation with SPION (item 2.8.1)

  1. If the PLL was used as a coating the authors must study the stability of this “functionalization”, determine the amount of PLL adsorbed in the SPIONS, and the possible changes in the size and charge of the SPIONS.

Answer:  Thank you for your suggestion. We performed a new experiment evaluating the hydrodynamic size distribution and zeta potential of different colloidal solutions using in the cellular labeling, and one the evaluated conditions is the presence of PLL, being performed the  size and charge measures of the SPIONs

  1. A similar concern arises regarding the filter conditions in cell studies. The SPIONS were submitted to filtration before the studies? If so, how this affects the administered dose? It would be important to present DLS size distribution graphics before and after filtration.

Answer:  Thank you for your observation Six different conditions of colloidal suspension containing SPION were used in this study, among these conditions there were those filtered and without filtration. The samples that were filtered contain a lower concentration of SPION due to the elimination of the second peak with a mean diameter close to 2.5µM as shown in Figure 5 through DLS measurements before and after filtering.

  1. Figure 4, scale bars are missing.

Answer:  Thank you for your observation. We added in the legend the information that all images used 10x magnification.

  1. Figure 6, the graphics A-H are difficult to read. Which concentration of SPIONS were used in this study?

Answer:  Thank you for your observation. We have modified the figure to improve the information understanding as well the concentration, which was 100 µgFe/mL, information added in the text and in the caption.

  1. Figure 7, cytometry data shows a bad compensation/calibration of PI and Annexin V channels.

Answer:  Thank you for your observation. We performed a new compensation adjustment for PI and Annexin V channels, for that, we have used the single stained tubes for Annexin-V and the single stained tube for Propidium iodide as you can see in the dot plots inserted below there are no spilover between the channels, the new analysis  were added to Figure 8.

 (See figure in the WORD file) 

  1. The authors state that the PLL increase the SPIONS uptake by increasing its positive charge, but how the poly-L-lysine interacts with the SPIONS to mediate its uptake on the cells?

Answer:  Thank you for your suggestion. We have added more information about the possible effects of PLL on the SPION charge in the discussion section of the manuscript.

Round 2

Reviewer 1 Report

Accept. The authors have made substantial changes addressing the comments raised by the reviewer.

Author Response

Reviewer #1

Accept. The authors have made substantial changes addressing the comments raised by the reviewer.

Answer: Thank you for your considerations.  We strive to carry out the proposed changes in the review.

Reviewer 3 Report

The authors addressed most of the reviewers concerns. The manuscript can now be considered for publication. 

Nevertheless, some proofreading must be performed (e.g., "SPIONAmine suspense in RPMI had an increase"). 

Additional discussion must be added to the text, namely the impact of the nanomaterials' aggregation (data shown in the colloidal stability studies)  on their in vivo performance.

Author Response

Reviewer #3

The authors addressed most of the reviewers concerns. The manuscript can now be considered for publication.

Nevertheless, some proofreading must be performed (e.g., "SPIONAmine suspense in RPMI had an increase").

Answer: Thanks for your suggestion. We proofreading all manuscript as requested.

Additional discussion must be added to the text, namely the impact of the nanomaterials' aggregation (data shown in the colloidal stability studies)  on their in vivo performance.

Answer: Thanks for your suggestion. We added more information in the discussion section of the manuscript as requested
